# PHYSICS-BASED SKINNED DANCE GENERATION WITH RL FINE-TUNING

## ABSTRACT

Denoising Diffusion Probabilistic Models (DDPMs) have shown great potential in generating high-fidelity, diverse, natural dances consistent with given music. However, due to the scarcity of skinned human motion data and the complexity of mesh data, existing methods mainly focus on generating dance moves in the form of skeletons, overlooking the domain gap between the skeletal structure and the human body geometry. When skeletal motions are visualized with human body mesh, anomalies such as torso interpenetration and imbalanced movements become highly noticeable. This physical implausibility significantly diminishes the aesthetic appeal of the generated dances and hinders their practicality in real-world applications. To address this issue, we propose a physical reward to fine-tune the diffusion model. Specifically, We first train a motion imitation policy in a physical simulator and use it to evaluate the physical plausibility (e.g., penetration, foot sliding) of generated motions. Ideally, generated motions that are more physically plausible will be easier to imitate, which means higher rewards. So we fine-tune the diffusion model to generate more physically plausible motions through Reinforcement Learning Fine-Tuning (RLFT). Furthermore, we find that the physical reward tends to push the model to generate freezing motions for less torso intersections. To mitigate it, we proposed an anti-freezing reward to balance the preference for freezing motions. Experiments on the human dance dataset show that our method can significantly improve the physical plausibility of generated motions, thereby generating dances that are aesthetically pleasing and realistic.

## 1 INTRODUCTION

Dance is a universal art form for humans to convey emotions, spread messages, and express their thoughts (LaMothe, 2019; Zambrano, 2023). Therefore, many applications such as film production, game development, and virtual reality experiences have the need for dance generation or dance animation. However, creating new choreography from scratch or capturing human dance movements through motion capture is not only costly but also time-consuming. Recently, deep-learning-based generative models have shown great potential in the art generation area, shading light into the music-conditioned dance generation. Despite the complexity of skeletal movements in human dances and their intricate relationship with the music condition, recent methods (Li et al., 2021; Siyao et al., 2022; Tseng et al., 2023; Sun et al., 2022) have made significant progress in generating high-quality, natural, and diverse dances that align well with the given music.

However, due to the complexity of representing and training with body meshes, most existing methods overlook the skinned mesh when generating dance data. Instead, they typically represent actions through bone rotations in skeleton form. However, the skeletal structure is only the intermediate result. When skeletal actions are visualized as body meshes, many physically implausible phenomena might occur, such as body interpenetration and imbalanced movement. These issues significantly degrade the aesthetic appeal and realism of the final visual results (Hoyet et al., 2012).

We argue that the problem lies in the domain gap between the learned skeletal structure and the human body geometry. To tackle this problem, we aim to incorporate physical constraints imposed by body meshes into the learning process of the generative model. Specifically, we first train an imitation policy using imitation learning on expert datasets AMASS (Mahmood et al., 2019) and AIST++ (Li et al., 2021), following prior works (Yuan et al., 2023; Yuan & Kitani, 2020), in a phys-

ical simulator (IssacGYM (Makoviychuk et al., 2021)). The imitation policy learns to mimic an input action in the physical simulator, forcing the output action to comply with the physical laws. Then, we construct a reward, called imitation reward, as a physical-aware signal for fine-tuning the diffusion model through reinforcement learning (RL) as proposed in Black et al. (2024). The imitation reward is designed to evaluate the physical plausibility of the generated motion. Since actions with physical implausibility (*e.g.* with less penetration) are impossible for the imitation policy to imitate because of the constraints imposed by the physical simulator. Furthermore, such implausibility can hinder subsequent imitations or even result in the failure of the entire process. Consequently, these issues lead to the imitation policy receiving lower rewards. In this sense, the imitation reward can implicitly impose the skin-based physical constraints into the denoising diffusion process.

There is another way (Yuan et al., 2023) to inject physical constraint into the diffusion process, which is using the imitation policy as a physical-guided motion projection module during the inference process. Compared to this straightforward combination of the diffusion model and imitation policy, our method further fine-tunes the diffusion model. The benefits are two-fold: i) Directly adopting the imitated motion as the result may encounter issues such as jittering or even imitation failure. However, the results generated by the diffusion model can ensure the naturalness of the movements. ii) it is time-consuming to process the motion projection with the physical simulator, while our method can save the projection time.

During the training of the reinforcement learning, we find that the imitation reward tends to favor movements with small magnitude, which may encourage the diffusion model to generate freezing motions. Therefore, we propose an anti-freezing reward for the diffusion model to balance this preference for freezing motions. Specifically, we evaluate the magnitude of the generated motions by computing the velocity and acceleration of the pose and translation parameters. By combining the anti-freezing reward with the imitation reward, our method can mitigate the bias towards freezing motions and encourage large movements.

To validate our method, we leverage the state-of-the-art (SOTA) dance diffusion model EDGE (Tseng et al., 2023) for finetuning and evaluate the results on the AIST++ dataset (Li et al., 2021). Experiments show that the occurrence of physical implausibility has significantly decreased, such as body interpenetration and abnormal foot-ground contact. Several metrics are designed to measure these physical improvements quantitatively.

- We propose an RL training methodology for fine-tuning the diffusion model, encouraging the diffusion process to generate physically plausible dance motions. Through the careful design of imitation reward and anti-freezing reward, our method can correct physically implausible movements generated by a well-trained model, while preserving the original semantics of the dance movements to the greatest extent.

- We train an imitation policy through a physical simulator on expert datasets. After training, the imitation policy can serve as a physical-aware reward that can impose physical constraints, especially the one brought by skinned mesh, into the RL training.

- Our experimental results demonstrate a significant improvement in the physical plausibility of the generated dances, including penetration and foot-ground contact. The visual quality of the generated results has also been greatly enhanced in terms of realism and aesthetics.

## 2 RELATED WORK

### 2.1 HUMAN MOTION GENERATION AND MUSIC-TO-DANCE GENERATION

Generating realistic human motion has been extensively studied. Previous approaches (Lee et al., 2002; Kovar et al., 2002; Arikan & Forsyth, 2002) primarily rely on graph-based methods. They decompose motions into clips and then recombine them according to predefined principles. However, these methods struggle to generate diverse human motions, especially the dance that exhibits variations in speed, length, and tempos. This limitation arises from the reliance on fixed motion units and rigid composition rules. In recent years, with the emergence of deep learning and large-scale human motion datasets (Mahmood et al., 2019; Li et al., 2021), numerous works have explored the use of various neural networks to generate diverse human motion (Tevet et al., 2023; Jiang et al., 2023; Liang et al., 2024; Dai et al., 2024). For instance, in the domain of music-to-dance generation, re-

cent methods utilize various network structures, including CNNs (Holden et al., 2016), RNNs (Tang et al., 2018; Yalta et al., 2019; Alemi et al., 2017; Huang et al., 2023), GCNs (Yan et al., 2019; Ren et al., 2020; Ferreira et al., 2021), GANs (Lee et al., 2019; Sun et al., 2020), Transformers (Li et al., 2023; 2020; 2021; Siyao et al., 2022) and Diffusion Models (Tseng et al., 2023; Alexanderson et al., 2023; Li et al., 2024), to better capture the intricate relationship between dance joint movements and accompanying music. However, most existing methods focus on improving the synchronization between music and human joint movements without considering constraints from the laws of physics (*e.g.* skin collision and gravity). Therefore, these methods tend to generate physically implausible motions. On the contrary, our method employs a physical simulator to model the laws of physics and instills physical knowledge into the diffusion model.

## 2.2 PHYSICS-BASED HUMAN MOTION MODELING

Physics-based human motion imitation is first utilized to generate realistic and controllable locomotion for characters in the physical simulator (Liu & Hodgins, 2017; Yuan & Kitani, 2020; Liu & Hodgins, 2018; Peng et al., 2018; Wang et al., 2017; Merel et al., 2017; Won et al., 2020; Park et al., 2019; Bergamin et al., 2019). Recent advancements have also adopted physics-based human motion imitation for more downstream tasks such as 3D human pose estimation (Zell et al., 2017; Rempe et al., 2020; Shimada et al., 2020; 2021; Yuan & Kitani, 2018; 2019; Isogawa et al., 2020; Yi et al., 2022; Yuan et al., 2021; Luo et al., 2022b;a) and 3D human motion generation (Yuan et al., 2023; Yao et al., 2023; Gillman et al., 2024). These studies leverage physics-based human motion imitation during inference to convert the generated motion to the physically plausible motion. However, many of them heavily depend on the ability of the imitation policy. There are chances that the imitation policy may fail to accurately imitate certain motions. Moreover, the imitation process in the post-processing will be time-consuming. Unlike previous work, we propose to incorporate physical laws into the motion generation model through RL fine-tuning. As a result, the generative network can be trained to directly synthesize the physics-aware motion without post-processing, thereby bypassing the inherent defects associated with the imitation policy.

## 2.3 REINFORCEMENT LEARNING FINE-TUNING OF DIFFUSION MODELS

With the success of fine-tuning large language models (LLMs) with reinforcement learning (Bai et al., 2022b;a; Lee et al., 2023a; Ouyang et al., 2022), recent research has proposed RLFT algorithms for text-to-image diffusion models. RWR (Lee et al., 2023b) firstly introduces the human feedback reward to fine-tune diffusion models. However, RWR ignores the sequential nature of the denoising process. Therefore, DDPO (Black et al., 2024) and DPOK (Fan et al., 2023) treat the denoising process of the diffusion model as a multi-step Markov Decision Process (MDP). Concurrently, Diffusion-DPO (Wallace et al., 2023) modifies the Direct Preference Optimization (DPO) algorithm (Rafailov et al., 2023) to directly optimize diffusion models based on preference data.

In contrast to previous methods that primarily relied on data-driven learning or heuristic constraints, our approach uniquely incorporates physics-based human motion imitation. By leveraging RLFT, we ensure that the physical limitations of human motion are instilled into the diffusion models, offering a more robust integration of physical constraints than previous approaches.

## 3 METHOD

As overviewed in Fig. 1, our method adopts the RL strategy which can instill the learned physical constraint into the dance diffusion model. Firstly, an imitation policy that can effectively mimic the dancing sequence in the simulator is trained on an expert dataset AIST++ (Li et al., 2021) (illustrated in Sec. 3.1). This well-trained imitation policy can then act as a reward evaluator, assessing the physical plausibility of the generated motion. Secondly, for training the dance diffusion model, we treat the denoising process as a Markov Decision Process (MDP). This allows us to employ multiple rewards to fulfill a reinforcement learning fine-tuning strategy, including the imitation reward provided by the imitation policy. (illustrated in Sec. 3.2).

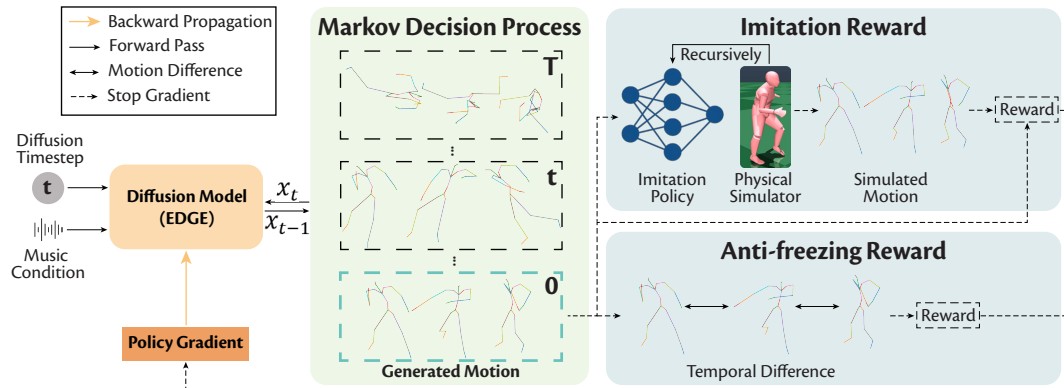

Figure 1: The overview of our method. Our method formulates the denoising process as a multi-step Markov Decision Process, allowing the diffusion model to be fine-tuned through Reinforcement Learning. To incorporate physical constraints into the diffusion model, we introduce an imitation reward based on physics-driven human motion imitation, which can evaluate the physical plausibility of the motion. Additionally, we design an anti-freezing reward to mitigate the imitation reward's preference for freezing motions.

### 3.1 IMITATION POLICY FOR IMITATION REWARD

To enable the diffusion model to learn the physical constraints of the real world, we need a metric to evaluate whether the generated motion obeys the physical laws. Inspired by Yuan et al. (2023), we found that an imitation policy can effectively serve this purpose. Human motion imitation policy is commonly used in robotics to control agents in the physical simulator to replicate complex movements. Since the motions replicated in the simulator inherently satisfy the various physical laws we set for the simulator (such as gravity, object collision, friction, etc.), we argue that the more the original motion conforms to physical laws, the more accurately it should be replicated in the physics simulator. Therefore, to utilize this characteristic during the training of the human motion diffusion model, we train an imitation policy to control the character with SMPL skinned mesh (Loper et al., 2023), which inherently obeys physical constraints such as gravity and human-body collisions.

We formulate human motion imitation as an MDP (Yuan & Kitani, 2020; Peng et al., 2018). The overall process can be defined using states (s), actions (a), transition dynamics ($\mathcal{T}$), reward functions (r), and a discount factor ($\gamma$) as follows,

$$s_t = \left(S_t, x_{t+1}^{res}, \psi\right), \qquad a_t = x_{t+1}^A, \qquad s_{t+1} = \mathcal{T}\left(s_{t+1}|s_t, a_t, \Phi\right),$$

$$\pi_\theta\left(a_t|s_t\right) = \mathcal{N}(\mu_\theta(s_t), \Sigma), \quad r\left(s_t, a_t\right) = \sum_{s \in S} w_s \exp\left(-\alpha_s \left\|x_t^s - \overline{x}_t^s\right\|^2\right) \tag{1}$$

$$+ w_r \exp\left(-\alpha_r \|F_r\|^2\right),$$

where $S_t$ represents the state of the controlled character, including joint angles, joint velocities, rigid bodies' positions, rotations, and linear and angular velocities. $x_{t+1}^{res}$ is the residual between the current state of the character and the next frame of the reference motion. $\psi$ corresponds to the SMPL (Loper et al., 2023) parameters of the character. In each step, the imitation policy $\pi_\theta$ takes action based on the given state to control the character to mimic the next pose of the given reference motion in the physical simulator. Subsequently, the character's state transits based on the transition dynamics and we can extract the imitated motion from it. Finally, we calculate the reward reflecting how well the reference motion is imitated, which can be used for reinforcement learning. Here, we mainly illustrate the components of Actions, Transition, and Rewards, for more details on other components, please refer to Yuan et al. (2023) for a comprehensive discussion.

**Actions** The policy outputs an action $x_{t+1}^A$, which represents the joint angles of the target pose for the character. A PD controller is then applied to drive the character towards this target pose. Additionally, to improve the imitation ability, we incorporate a residual force (Yuan & Kitani, 2020) applied to the character's pelvis. This extra force helps to stabilize the character during movement.

**Transition** $\mathcal{T}$ is the transition dynamics of the simulator and $\Phi$ denotes the physical constraints modeled by the physical simulator, including gravity, body collision, skeletal structure, etc. Notably, unlike previous methods (Yuan et al., 2023; Yao et al., 2023; Gillman et al., 2024), we further integrate body collision handling into both the training and inference stages, allowing the imitation policy to better replicate the reference motion under collision constraints. Specifically, we treat each part of the SMPL model's mesh (Loper et al., 2023) as the collision volume and disable collision checks between adjacent joints to prevent neighboring joints from being stuck and unable to move.

**Rewards** The reward consists of two components. The first part is designed to encourage the imitated motion to match the ground truth. S includes components representing human motion: local joint rotations, joint velocities, 3D world joint positions, and global joint rotations. Additionally, $w_s$ and $\alpha_s$ are the weighting factors of each component of the rewards. Finally, $x_t^s$ is the t-th frame of the imitated motions represented in s, which is extracted from $s_t$, and $\overline{x}_t^s$ is the t-th frame of the ground truth in the training set.

The second part of the reward is a regularization term that limits the magnitude of the residual force, as excessively large residual forces can harm the physical realism of the character. $\alpha_r$ and $w_r$ are weighting factors, and $F_r$ is the residual force.

**Training Strategy** As the MDP has been defined, we can optimize the imitation policy using an RL algorithm (Schulman et al., 2017). To enhance the imitation policy's ability to mimic dance, we first train the imitation policy on the AMASS dataset (Mahmood et al., 2019) which contains a wide variety of general human motions, and then fine-tune the imitation policy on the AIST++ dataset (Li et al., 2021) which is a commonly used dance dataset.

## 3.2 PHYSICS-BASED DANCE DIFFUSION MODEL

With the well-trained physics-based imitation policy, we conduct an RLFT strategy to distill the physical constraints (body geometry, gravity, friction, etc.) of the simulator into the diffusion model.

**RLFT Formulation** To conduct RL fine-tuning on diffusion models, recent works (Black et al., 2024; Fan et al., 2023) proposed to formulate the diffusion denoising process as a multi-step MDP:

$$s_t = (c, T - t, x_{T-t}), \qquad a_t = x_{T-(t+1)}, \qquad r(s_t, a_t) = \begin{cases} r(x_0, c) & \text{if } t = 0, \\ 0 & \text{otherwise.} \end{cases}, \qquad (2)$$

$$\pi(a_t|s_t) = p_\theta\left(x_{T-(t+1)}|c, T-t, x_{T-t}\right), \qquad r(x_0, c) = r_{imit}(x_0) + r_{anti}(x_0),$$

where $t$ denotes the $t$-th decision step, while $T$ is the total number of steps. $c$ is the condition signal (*i.e.* music sequence in our task). $x_{T-t}$ is the denoised motion at denoising step $T - t$. $p_\theta$ is the diffusion model being fine-tuned. The reward consists of two components: $r_{imit}$, which evaluates the physical plausibility of the generated motion, and $r_{anti}$, which is designed to eliminate freezing issues, as we will illustrate later. Then we can use any policy-based RL algorithm, such as REINFORCE (Williams, 1992), to optimize the diffusion model based on the task-oriented rewards. To ensure training stability, we further adopt a pure on-policy training strategy. Different from the off-policy training or training with multi-step updates, which may leverage the data collected by older policies for better sample efficiency, we only use the data collected by the newest policy to update itself. Meanwhile, to enhance the numerical stability and convergence, we also normalize the reward to have zero mean and unit variance as in Black et al. (2024). The reward's mean and standard deviation statistics are tracked for each music independently:

$$A_i(x_0, c) = \frac{r_i(x_0, c) - \mu(c)}{\sigma(c)} \quad i \in \{imit, anti\}. \qquad (3)$$

After collecting enough trajectories, we can update the diffusion model using the policy gradient function as follows:

$$\nabla_\theta J = E\left[\sum_{t=0}^{T} \nabla_\theta \log p_\theta\left(x_{T-(t+1)}|c, x_{T-t}\right) * (\alpha A_{\text{imit}}(x_0, c) + \beta A_{\text{anti}}(x_0, c))\right], \qquad (4)$$

where $p_\theta$ denotes the diffusion model being updated, and $A_{imit}(x_0, c)$ and $A_{anti}(x_0, c)$ are the normalized imitation and anti-freezing rewards for guiding the optimization, respectively. Additionally, $\alpha$ and $\beta$ are their respective weights.

**Imitation Reward**   As aforementioned, we leverage the learned imitation policy as an evaluator to determine whether the generated motion is physically plausible. Specifically, the imitation policy manages to mimic the generated motion in a physical simulator, yielding an imitated motion per generated motion. Ideally, if the generated motion obeys the real-world physical constraints, the imitated motion should be the exact same sequence as the generated motion. However, if the generated motion somehow violates the physical laws, such as an arm penetrating through the body, the imitation policy will attempt to produce a result that closely approximates the input motion while eliminating the interpenetration. In this case, there will be a difference between the generated motion and imitated motion. Therefore, the greater the difference between the two motions, the more physically implausible the generated motion is, as it becomes harder to imitate in the physics simulator. Consequently, we measure this difference and assign a larger reward to motions with smaller differences. The reward can then be formulated as follows:

$$r_{imit}(x_0, c) = \sum_{s \in S} w_s \exp\left(-\alpha_s \|x_0^s - \hat{x}_0^s\|_2\right), \tag{5}$$

where $S$ is a set in which each component is a specified representation of human motion, including local joint rotations, joint velocities, 3D world joint positions, and global joint rotations. $x_0^s$ is the generated motions represented in $s$, and $\hat{x}_0^s$ is the imitated motions. Additionally, $w_s$ and $\alpha_s$ are the weighting factors of each component of the rewards, and these values are kept the same as the training process of the imitation policy to ensure consistency in evaluating the difference between the reference and imitated motion.

**Anti-freezing Reward**   The imitation reward is designed to rate higher rewards for physically plausible motions, however, we find that freezing/slow-speed motions can also receive high rewards, as they are relatively easy to imitate. This phenomenon results in a tendency for the diffusion model to generate more freezing motions. To mitigate this bias, we propose an anti-freezing reward to encourage generating more dynamic motions. Specifically, we compute the velocity and acceleration of the motion from the pose sequence, and apply the mean square value as the anti-freezing reward:

$$r_{\text{anti}}(x_0, c) = \overline{v(x_0)^2} + \overline{a(x_0)^2}. \tag{6}$$

## 4 EXPERIMENTS

**Dataset**   We train the imitation policy and dance diffusion model with the AIST++ (Li et al., 2021) dataset, which is the most commonly used dataset in dance motion generation. It consists of 1,408 high-quality dance motions paired with music from a diverse set of genres. We follow the setup used in EDGE Tseng et al. (2023), the train/test splits are kept the same as the original dataset, and all the training examples are cut to 5 seconds, 30FPS.

**Implement Details**   For the imitation policy, we adopt Isaac Gym (Makoviychuk et al., 2021) as our physical simulator, in which we can detect the collision between the character's torsos. The weighting factors for calculating the reward ($w_s$ and $\alpha_s$) are set to (0.6, 0.1, 0.2, 0.1) and (60, 0.2, 100, 40) respectively. And $w_r$ and $\alpha_r$ are 0.1 and 30. For the diffusion model, we test our RLFT method on EDGE denoiser, and we employ denoising diffusion implicit models (DDIM) as proposed in Song et al. (2020) with 50 diffusion steps and classifier-free guidance (Ho & Salimans, 2022). In each fine-tuning iteration, we sample 2,048 motions from the training dataset of AIST++ (Li et al., 2021). We accumulate gradients across 50 denoising steps of all samples and perform one gradient update. Our optimizer adopts Adam (Kingma & Ba, 2017) optimizer, with learning rate set to 1e-6.

**Evaluation Metrics**   For evaluating our physically plausible dance generation results, there are two essential aspects: aesthetic quality and physical plausibility. For aesthetic quality, we conduct a user study ("Overall" in Tab. 1), and adopt several metrics from former works like beat alignment score (BAS) (Siyao et al., 2022) and $\text{Div}_k$ /$\text{Div}_g$ (Li et al., 2021). In terms of physical plausibility, the evaluation metrics include a user study ("Physical" in Tab. 1), Penetration Rate, physical foot contact score (PFC), and magnitude of motion.

In our user study, to highlight the performance of dance results on skinned mesh, we rendered all dance animations using the SMPL model (Loper et al., 2023). The metrics "Overall" and "Physical"

Table 1: Quantitative results on AIST++ (Li et al., 2021) dataset. The "Overall" and "Physical" columns represent the win rate of our method over the others, *i.e.* a higher value indicates better performance of our method. Note that FACT's low penetration rate mainly comes from its tendency to generate freezing motions, which is further proved by the magnitude of motion in Tab. 3.

| Method | Overall | Physical | Penetration Rate ↓ | PFC ↓ | BAS ↑ | $\text{Div}_k$ /$\text{Div}_g$ → |
|---|---|---|---|---|---|---|
| Ours | / | / | **90.14** | **0.9273** | **0.2897** | 2.93/2.14 |
| EDGE | 70% | 75% | 173.03 | 0.9523 | 0.2865 | 2.82/2.09 |
| FACT | 73.7% | 55.3% | 97.98* | 1.2125 | 0.2380 | 5.74/5.71 |
| Bailando | 86.5% | 78.4% | 176.36 | 1.5466 | 0.2320 | 7.83/6.33 |
| Ground Truth | 48.9% | 40% | 135.27 | 1.4699 | 0.2292 | 8.27/7.51 |

are the statistics of two questions: (1) "Which dance looked and felt better overall?" and (2) "Which dance is more physically plausible?". We conducted the survey through Prolific, a crowdsourcing platform, recruiting 40 participants for each pairwise comparison between methods. "BAS" aims to assess how well the generated dances synchronize with the music's beat. "$\text{Div}_k$ /$\text{Div}_g$" are the diversity of the generated dances within "kinetic" and "geometric" feature spaces. Meanwhile, we propose "Penetration Rate" to quantify the extent of body penetration. Specifically, it is defined as the average number of intersected faces in the body mesh per frame. Additionally, "PFC" is defined as the products of body root acceleration and the velocity of both feet (Tseng et al., 2023), which is used to evaluate the plausibility of foot-ground contact. Finally, the magnitude of motion is calculated as the average temporal difference of the pose across the entire sequence, which reflects the freezing extent of the generated motions. We argue that the magnitude of the motions should not be too low to prevent freezing motions.

## 4.1 EVALUATION ON THE SKINNED DANCE GENERATION

In this section, we compare our proposed method to several SOTA dance generation methods, including FACT (Li et al., 2021), Bailando (Siyao et al., 2022), and EDGE (Tseng et al., 2023). All the evaluation metrics are computed on 20-second dance clips for fairness.

**Human Perception Results** The "Overall" and "Physical" metrics in Tab. 1 represent the overall aesthetic quality and physical plausibility measured by human perception. Our method significantly surpasses other approaches, indicating a noticeable improvement in physical plausibility. On the other hand, our method also demonstrates improved results in overall aesthetic quality compared to EDGE (Tseng et al., 2023). Given that our dance sequence shares similar motion patterns with the pretrained model (*i.e.* EDGE), as shown in Fig. 2, this result shows that the physical plausibility can greatly affect the overall aesthetics in human perception. It is noteworthy that FACT (Li et al., 2021) also performs well in "Physical" rating, its "Overall" rating is inferior to ours. This is because they tend to generate motions with limited magnitude and slow speeds, which lacks aesthetic appeal. This point can be further proved by the magnitude of the motion. The magnitude of the motion from FACT is 0.4540, which is relatively small compared to our method (0.6877 in Tab. 3).

**Penetration rate** Penetration rate reflects the physical plausibility in terms of body part penetration. The results in Tab. 1 show that our proposed method achieves a significant improvement in this metric. Compared to the baseline model EDGE, we reduce the penetration rate by 48%. Also, FACT obtains a low penetration rate due to its tendency to generate small-magnitude motions, which is undesirable in aesthetics. We also provide quality comparisons between EDGE and our method in Fig. 2, where we can clearly see that after the RLFT training, our method has learned to replace previously implausible motions with similar yet non-interpenetrating ones. For instance, in the first and third rows of Fig. 2, our model carefully avoids interpenetration by increasing the amplitude of arm movements; in the second row, our model learned to avoid stepping on the supporting foot, thus preventing the body from tripping.

**Physical Foot Contact Score** Physical Foot Contact (PFC) score is used to evaluate the plausibility of foot-ground contact. As shown in Tab. 1, our method outperforms the other approaches in this

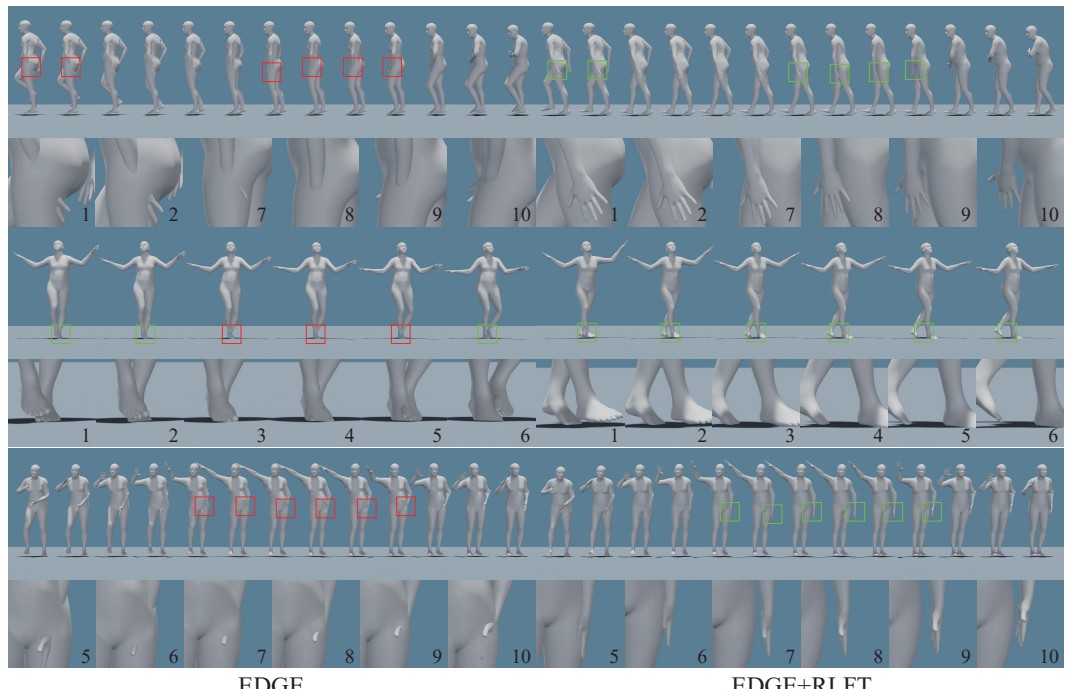

Figure 2: The visual comparisons of EDGE (Tseng et al., 2023) and our generated motions. Both motion sequences are generated with the same music and seed. Some body parts are enlarged for a better view. The red box signifies the presence of body penetration, while the green box indicates the improvement after the RLFT. The subscript number denotes the frame number.

metric, demonstrating that our approach effectively enhances the physical realism of foot-ground contact. We infer that the character in the physical simulator should obey Newton's law of motion, which is the basis of the definition of PFC.

**Other Non-physical Metrics** We also provide the results of other non-physical metrics in Tab. 1, to demonstrate our method will not deteriorate in other aesthetic performance, such as beat alignment (BAS) and diversity ($Dist_k$, $Dist_g$). Since our method is not targeted at solving the beat alignment and diversity issue, it maintains a comparable result with EDGE. However, it can be observed that diffusion-based methods, including EDGE and our proposed method, acquire a low score in diversity metrics. We infer that this is because of the way the diffusion model generates long sequences. It generates longer sequences by stitching short clips together. Then the diversity is calculated on the average of short clips. Therefore, while the diversity across individual short clips may be high, their composition can exhibit lower diversity.

## 4.2 ABLATION STUDY

Table 2: The ablation study on RL fine-tuning.

| Method | Penetration Rate ↓ | PFC ↓ |
|---|---|---|
| EDGE | 173.03 | 0.9523 |
| EDGE w/ Proj | 116.16 | 1.8801 |
| Ours | **90.14** | **0.9273** |

Table 3: The ablation study on the AF reward.

| Method | Magnitude→ |
|---|---|
| EDGE | 0.7188 |
| FACT | 0.4540 |
| Ours w/o AF | 0.5618 |
| Ours | **0.6877** |
| Ground Truth | 0.6735 |

We perform ablation studies on RL fine-tuning and the anti-freezing reward to validate the effectiveness of each component in our method.

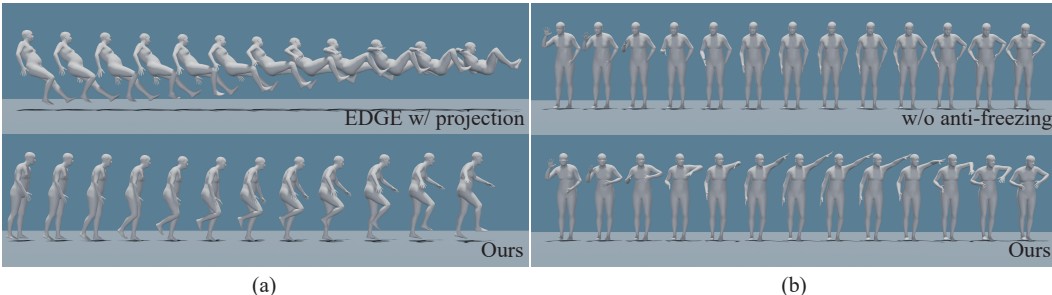

Figure 3: The visual comparison for ablation studies. Each compared dance pair is generated from the same audio track. (a) compares the results of directly projecting motion through a physical simulator with those of our proposed method. As shown, direct motion projection can result in falling motions due to the inability to accurately imitate the physical implausible movements. (b) presents an example result of the model trained without an anti-freezing reward. The model tends to generate small-amplitude movements without the anti-freezing reward.

**Ablation Study on RL Fine-tuning**   As another approach to leverage imitation policy, we can also apply the imitation policy as a post-processing step to the generated motions, similar to Yuan et al. (2023). Specifically, the imitation policy is used to mimic the motions generated by EDGE and output refined motions within the physical simulator. We refer to this method as "EDGE w/ projection". However, we identify some significant limitations with this method. First, the diffusion model may generate some motions that are physically impossible for the imitation policy to mimic directly, *e.g.* one leg passes through the support leg, which can lead to chaotic outcomes such as slipping or falling (As shown in Fig. 3(a)). The quantitative results in Tab. 2 also validate that although the penetration rate of "EDGE w/ Proj" has decreased, the PFC metric has significantly deteriorated. This may be due to abnormal foot motion, where the feet lose contact with the ground. In contrast, our method still works by assigning a very low reward for such "failling" scenarios. Moreover, besides a direct imitation that generates similar motion patterns, our method even gradually learns to avoid such hard cases by replacing them with an easier and physically plausible motion, which is also an advantage brought by RLFT. More video examples are provided in supplementary materials. Second, post-processing is time-consuming during inference Yuan et al. (2023), for the projection needs to be done in the physical simulator, and takes around 10 seconds on an NVIDIA A100 GPU.

**Ablation Study on Anti-freezing Reward**   To validate the effectiveness of the anti-freezing reward, we perform an ablation study evaluating the magnitude of motion in Tab. 3. As shown, compared to EDGE and our method, the magnitude of motion reduces significantly without anti-freezing reward (*i.e.* Ours w/o AF). Fig. 3(b) also shows the same conclusion that it tends to generate freezing motions without an anti-freezing reward, which is generally undesirable in practice.

## 5 CONCLUSION

In this paper, we propose a method that fixes the physical implausibility issue of the current dancing generation works through Reinforcement Learning Fine-Tuning (RLFT). Specifically, we utilize the physics-based human motion imitation policy as a reward to evaluate the physical plausibility of the generated motions. Through the physics-based reward, we can instill the physical law (*e.g.* body collision, gravity, and foot-ground contact), which was modeled by the imitation policy in the physical simulator, into the diffusion model through RLFT. Furthermore, to prevent the generation of freezing motions, we propose an anti-freezing reward to enhance the dynamic of the motions. We conduct extensive experiments to evaluate our method across multiple metrics, showing that it significantly improves physical plausibility, especially in terms of body penetration and foot-ground contact. We believe our approach has the potential to advance the real-world applications of motion generation. In the future, we aim to further refine both the physical plausibility and smoothness of the generated motions for direct use in downstream applications.

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
