# OpenReview forum: "Physics-based Skinned Dance Generation with RL Fine-tuning"
_ICLR.cc/2025/Conference — ICLR 2025 Conference Withdrawn Submission_

### Official Review · Reviewer_2ZqM · 2024-10-20

**Soundness:** 2
**Presentation:** 3
**Contribution:** 2
**Rating:** 5
**Confidence:** 4

**Summary:**

The paper aims to propose a better physically-appealing human dancing imitation learning method. Specifically, imitation reward and anti-freezing reward are proposed for RL, contributing to encouraging imitation quality and reducing the possibility of motions with small magnitude, respectively.

**Strengths:**

1. The paper proposed two rewards targeting to a more physically-appealing motion imitation learning. The corresponding heuristics e.g. motion magnitute make sense in most cases.
2. The proposed RL training methodology is well-suited for capturing and imitating various motion sequences with large diversity.

**Weaknesses:**

1. L191-L193 demonstrated the importance of controling the character with SMPL skinned mesh because it is a physically-constrainted model, while L192-L222 claimed that disabling the collision checks between adjacent joints in both training and inference helps better replicating the reference motion under collision constraints. This is a bit confusing to me because physically constraints inherited
in SMPL are important for motion imitation/synthesis and they are also (I believe) the motivation of the paper itself by involving the step: physical simulator->simulated motion as shown in Fig1.
2. The proposed method trained its imitation policy on AMASS dataset and then finetuned on AIST++, which is a different setting as EDGE. So the comparisons between the two methods seems not fair to me as AMASS comprises many realistic human motions that can apparently help for imitation policy training without any other explict physical constraints.
3. It seems that an optimization based 'projection' step from the generated motions to physically-appealing AMASS dataset as [1,2] can also induce a motion refinement as post-processing, which is very simple and more straightforward to distill the prior of large human motion datasets. While the proposed  Anti-freezing Reward is heuristic-based and the generated motions as physical simulator is not the same as the extracted simulated motions.

[1] Tiwari et al. Pose-NDF: Modeling Human Pose Manifolds with Neural Distance Fields. ECCV 2022. \
[2] NRDF: Neural Riemannian Distance Fields for Learning Articulated Pose Priors. CVPR 2024.

**Questions:**

1. The ablation about w/wo collision checks in both training and inference is expected and why the proposed method can generate motions with less torso intersections even without collision checks?
2. The ablation about training imitation policy on only AIST++ v.s. the training scheme as in the paper is expected.

---

### Official Review · Reviewer_p3zx · 2024-10-22

**Soundness:** 4
**Presentation:** 4
**Contribution:** 3
**Rating:** 6
**Confidence:** 4

**Summary:**

This paper introduces an approach for generating physically feasible human motion from music input. The method uses a reinforcement learning-trained imitation policy to fine-tune a diffusion model. The fine-tuning process is formulated as a reinforcement learning problem, where an imitation reward and a non-freezing reward are introduced. The imitation reward guides the generation toward physically feasible motion, while the non-freezing reward prevents the generation of stationary motions. The proposed approach is compared with several baselines in terms of overall motion quality and physical feasibility. The results indicate that the proposed method outperforms baseline methods in terms of physical feasibility. The supplementary videos provide visualizations of the generated motions.

**Strengths:**

- The overall framework is simple and practical.
- Different baseline methods have been compared, and the results have been analyzed thoroughly.
- The related work is sufficiently covered.

**Weaknesses:**

- In Table 1, the Human Perception Results for the proposed method are marked as ‘/’.
- I believe the results are highly dependent on the performance of the imitation policy and the simulation settings. It would be valuable to see some analysis in this direction.
- In the ‘Other Non-physical Metrics’ section, the authors claim that the low diversity issue is caused by short motion clips. Would it be possible to compare the diversity of different methods and the ground truth by clipping the motions to the same length as the diffusion model?

**Questions:**

Please check the weakness part.

---

### Official Review · Reviewer_NAXB · 2024-11-03

**Soundness:** 3
**Presentation:** 2
**Contribution:** 2
**Rating:** 3
**Confidence:** 4

**Summary:**

This paper proposes reinforcement learning-based finetuning of diffusion models for music-to-dance generation to overcome problems with physical plausibility of generated movements. An imitation policy is trained to mimic real-world ground truth movements, resulting in a reward function later used for the finetuning of the generator. To avoid trivial static movements, an additional anti-freeze reward function is also proposed. The proposed method builds on that of Yuan et al 2023, but instead of using the imitation policy for projection for generated movements to become physically plausible, the policy is used for finetuning the diffusion model using an MDP-based formulation of the training process.

**Strengths:**

The idea to first train an imitation policy to mimic the physical world and then use the policy to replace a physical simulator to finetune a diffusion model toward generating physically plausible motions is innovative indeed. The feasibility of the proposed framework is also demonstrated in the experiments, where it is evaluated and compared against three state-of-the-art methods and is shown to be superior in terms of both aesthetics and physical plausibility using both quantitative and qualitative measures.

**Weaknesses:**

The proposed work borrows a lot respectively from Yuan et al 2023 and Black et al 2024, both in terms of the underlying ideas and the formulations. There is nothing wrong with this if the result is good. However, the way the proposed method is described is far from clear. Possibly due to the lack of space, a lot is left to be implicitly understood, instead of explicitly explained.

The paper first describes the training of an imitation policy and later the reinforcement learning-based finetuning that includes an imitation reward function. However, the link between the two is unclear. How do you go from the policy in (1) to the reward function in (5)? Is the policy used to produce a sequence, which is later used in the reward function? But since the policy is stochastic, is only the mean used? The description of the imitation reward is also confusing since it talks about generated motion in a physical simulator when the generated motion ought to come from the diffusion model. By the way, what depends on $c$ in (5)?

In the notations, many different things are referred to as states or poses: $S_t, s_t, x_{t}^s, x_{t+1}^A, x_{t+1}^{res}, \bar{x}_t^s, x_t, x_0^s$, and $\bar{x}_0^s$. These are not fully described. For example, what does “residual between the current state of the character and the next frame of the reference motion” mean? Which state is referred to and what “reference motion”? No such reference motion has yet been mentioned. Also confusing is that $t$ is used to both denote time and diffusion step. The $0$ in $x_0^s$ should be understood as diffusion step $0$, but $t$ in $x_t^s$ refers to time $t$. Do $x_0$ and $x_t$ in (2) refer to full sequences, while $x_t^s$ in (1) only includes one time-step? Using $s$ in the notation is also confusing since $s$ is often used to indicate the target diffusion step in papers about diffusion, while here it is something else.

While other parts of the paper are easy to read and understand, the theoretical parts ought to be completely rewritten. The way it is written now seems a bit sloppy with derivations more or less directly following those in Yuan et al 2023 and Black et al 2024, but without updating the notations so that everything can be fully integrated and understood.

**Questions:**

* Other than the evaluation, what makes the proposed system specific for dance generation?
* What is the reason why freezing sometimes occurs when there is no anti-freeze reward function?
* Why are the numbers exactly 70% and 75% for EDGE in Table 1? How many sequences are evaluated?
* Did the 40 participants know more about dance than others? How were they selected?

**Details Of Ethics Concerns:**

There are no concerns.

---

### Official Review · Reviewer_1U9z · 2024-11-04

**Soundness:** 3
**Presentation:** 3
**Contribution:** 2
**Rating:** 5
**Confidence:** 4

**Summary:**

This paper provides a reinforcement learning (RL) fine-tuning strategy for a diffusion model aimed at dance generation, helping the model avoid (1) freezing during dancing and (2) penetration. Specifically, a policy maker is first trained to imitate dance motion in a physical simulation; then is is fine-tuned using policy gradients in an on-policy manner with both anti-freezing and imitation rewards. Experiments show that this proposed strategy indeed improves physical plausibility and prevents freezing as expected.

One potential shortcoming of this submission is the lack of any comparison to existing methods of physical plausibility (like PhysDiff). Meanwhile, there exists some unclear description/statements.

**Strengths:**

+ Physical plausibility in dancing is a significant problem that has not yet received much attention.

+ Experiments, particularly the demos, show that it successfully avoids penetration and freezing.

+ The writing is generally easy to follow.

**Weaknesses:**

- What is the difference between the proposed RL fine-tuning (RLFT) and existing methods like PhysDiff? Specifically, what is the relationship between the imitation policy part (Section 3.1) and PhysDiff? I also think it would be beneficial to include a comparison with PhysDiff in the experiments.

- The details of the on-policy RLFT process may be somewhat unclear. For example, it is stated, "we sample 2,048 motions from the training dataset of AIST++." This is a bit ambiguous. Since this is RL fine-tuning, the samplings are generated by the current policy, while as I understand, the diffusion model (EDGE) is already well-trained. Can samplings from the training set provide a balanced input, including negative aspects (penetration/freezing)?

- In the ablation study, I find "EDGE w proj" somewhat confusing; it is described as "similar to PhysDiff" (Line 451). Why do the authors consider PhysDiff as "post-processing"? As I understand it, PhysDiff embeds the simulator into the diffusion models within the final few iterations, which doesn’t seem all equivalent to "post-processing." Could the authors clarify the implementation of "EDGE w proj"? Also, in the demo, the "post-processing" approach results in floating in the air. This seems odd and confusing, as a physical simulator is expected to prevent this.

- Some related works are missed, e.g., Bailando++ (Siyao et al., 2023).

**Questions:**

See weakness.

---

### Note · Authors · 2024-11-15

I have read and agree with the venue's withdrawal policy on behalf of myself and my co-authors.